# Study protocol: a mixed-methods realist evaluation of the Universal Health Visiting Pathway in Scotland

Lawrence Doi  ,[1] Kathleen Morrison,[1] Ruth Astbury,[2] Jane Eunson,[3] Margaret A Horne,[4] Ruth Jepson,[1] Louise Marryat  ,[5] Rachel Ormston,[3] Rachael Wood  [6,7]

► Prepublication history and additional material is published online only. To view please visit the journal online (http://dx.doi.org/10.1136/bmjopen-2020-042305).

¹Scottish Collaboration for Public Health Research and Policy, School of Health in Social Science, The University of Edinburgh, Edinburgh, UK
²School of Health and Life Science, University of the West of Scotland, Paisley, UK
³Ipsos MORI Scotland, Edinburgh, UK
⁴Centre for Population Health Sciences, The University of Edinburgh, Edinburgh, UK
⁵Salvensen Mindroom Research Centre, The University of Edinburgh, Edinburgh, UK
⁶Information Services Division, NHS Scotland National Services Division, Edinburgh, UK
⁷Child Life and Health, The University of Edinburgh, Edinburgh, UK

**Correspondence to**
Dr Lawrence Doi;
larry.doi@ed.ac.uk

## ABSTRACT

**Introduction** The growing political emphasis on the early years reflects the importance of these formative years of life. Health visitors in the UK are uniquely positioned to improve health outcomes for children and families and to reduce health inequalities. Recently, there has been a policy change in Scotland in an attempt to enhance the delivery of the universal health visiting service. This study aims to examine the extent to which the enhanced Universal Health Visiting Pathway is implemented and delivered across Scotland and to assess any associated impacts.

**Methods and analysis** A mixed-methods study incorporating four methodological components and uses realist evaluation as the overall conceptual framework. It comprises three phases (1) initial programme theory development; (2) programme theory validation and (3) programme theory refinement. The programme theory validation will use interview and focus group data of parents and health visitors, and conduct a case note review at five study sites. It also involves a national survey of parents and health visitors and routine data analysis of existing secondary data. The analyses of the ensuing qualitative and quantitative data will be carried out using a convergent mixed-methods approach to ensure continuous triangulation of multiple data. The findings of the evaluation will provide contextually relevant understanding of how the Universal Health Visiting Pathway works and evidence the impact of increased investments in health visiting in Scotland.

**Ethics and dissemination** This protocol has been approved by the School of Health in Social Science Research Ethics Committee, University of Edinburgh. Additional approvals have been granted/will be sought from the Public Benefit and Privacy Panel for health and social care in Scotland for the case note review, survey and routine data analysis elements of the evaluation. The findings will be prepared as reports to the funders and presented at conferences. It will be submitted for publication in peer-reviewed journals.

## Strengths and limitations of this study

► This is a large study using a theory-driven approach of realist evaluation to examine the implementation and delivery of a national Universal Health Visiting Pathway.

► The study is expected to generate contextually relevant understanding of how the Universal Health Visiting Pathway works and evidence the impact of increased investments in the programme, while identifying areas for improvement, which may be relevant to other similar programmes.

► Using mixed-methods ensures that both qualitative and quantitative data are employed to provide continuous triangulation of data, thereby enhancing the credibility of the study findings.

► By not using well-established experimental research approaches, some may argue against the robustness of the study design.

and well-being, education, achievement and economic status.[1] Recent early year policies across the UK reflect this and have emphasised the need for greater prevention, early identification and intervention during the early-years stage (prebirth to 5 years old). The early years, including child development, protection and welfare, have been recognised as key public health priorities, which are crucial to reducing health inequalities across the life course.[2] Additionally, children who experience disadvantage in the early years are at a higher risk of injury, social, emotional and cognitive difficulties.[3]

Health visitors have played a longstanding and vital role supporting children and families throughout the UK. Health visiting practice has been firmly rooted in public health for over 150 years, with overall goals of health promotion and disease prevention. The professional practice of health visiting comprises 'planned activities which aim to improve the physical, mental, emotional and social health and well-being of the population,

## INTRODUCTION

The formative early years, which lay the foundations for physical, intellectual and emotional development, have profound life-long implications for individual health

preventing disease and reducing inequalities in health'.[4] Health visiting is a proactive service that searches for, and responds to, health needs at the individual, family, group and community level.

Since its inception, the health visiting profession has been heavily influenced by the policies of changing governments. Over the years, the pushing and pulling of various agendas have drastically altered the scope of the profession and the nature of the service it provides.[5] This is in part due to the fact that health visitors work with children, families and communities, areas where strong and often polarising political views are held. More so, health visitors are uniquely positioned to work across health and social care divides, which few other professions are able to do. This enables them to deliver a range of health promotion, intervention and illness-prevention activities to individuals and families in a variety of settings.

The growing recognition for the importance of the early years, the current political emphasis on this and a growing evidence base for health visiting practice has led to large reinvestments in health visiting services across the UK. Government policies have sought to harness health visiting as a key asset to improve health outcomes for children under the age of 5 and to reduce health inequalities. Efforts have also been made nationally to recruit, train and employ more health visitors leading to increased recognition and redefinition of the health visiting profession as a whole.

In 2013, the Chief Nursing Officer's Directorate, Scottish Government, undertook a scoping exercise of health visiting practice in Scotland. The findings demonstrated that there was a significant degree of variation across the service in terms of assessment, resources and visiting patterns being delivered by health visitors to families in Scotland. A refocused approach to health visiting was published by the Scottish Government in 2013.[6] The changes took into account the changing policy landscape relating to the early years, children and families, and sought to ensure that workforce capability and capacity would be equipped to successfully deliver these policies. Following substantial investment in the service, the Universal Health Visiting Pathway (UHVP) was introduced in 2015.

The UHVP refocuses the role of the health visitor and includes changes to caseload weighting and management; intervention delivery; education, training and resources; and visiting patterns. The aim of this study is to examine the extent to which the UHVP is implemented and delivered across Scotland and to assess any associated impacts. To achieve this, a robust mixed-methods realist evaluation proposal has been developed to understand 'what works for whom, why and in what circumstances'. Incorporating four methodological components, this paper presents a protocol describing the use and purpose of these methods over two stages of a national-level evaluation project.

## METHODS AND ANALYSIS
### Conceptual framework

This is a mixed-methods study that uses realist evaluation as the overall conceptual framework to examine the UHVP across Scotland. Realist evaluation is increasingly used to evaluate nursing and healthcare programmes because of the valuable insight it provides into how such programmes work and how they can be improved.[7] Realist evaluations are well suited to addressing the complexity of healthcare systems and are able to produce useful findings for decision makers keen to improve healthcare programmes.[8] The strength of realist evaluation is that it can draw valuable lessons about how particular conditions make particular outcomes more likely.[9] Its goal is to explain and generate knowledge of how to improve a programme. It can also provide transferable lessons that may be used by others who intend to implement a similar programme elsewhere.[10] Realist evaluation argues that in order for the findings of evaluation to be useful to decision makers, there is the need to ask the question 'what works for whom, why and in what circumstances?' By asking this question, the evaluation focuses on explaining how outcomes are achieved in respect to their contexts. This is particularly important because the UHVP has been implemented in the midst of significant policy developments in maternal and child health within Scotland, including the Family Nurse Partnership programme[11] and the Children and Young People (Scotland),[12] which includes Getting it Right for Every Child and Named Person policies.[12] It is important that outcomes of the evaluation are sensitive to such policy contexts.

The study will evaluate the contexts, mechanisms and outcomes of the UHVP, and make recommendations for the longer term sustainability of the service in Scotland. Realist evaluation is well suited to evaluating complex interventions, allowing for a theoretical and methodologically rigorous approach. In order to ascertain what works and understand how the UHVP improves the health and well-being of children and families in Scotland, there is the need to interrogate how contexts interact with causal forces, powers and processes that generate change. In line with a realist evaluation approach, and as outlined in figure 1, this study comprises three phases: (1) initial programme theory development; (2) programme theory validation; and (3) programme theory refinement and the development of lessons learnt. Realist evaluation is methods-neutral and the use of a mixed-methods approach employing a case note review, interviews and focus groups with parents and staff, as well as parent and staff surveys, and analysis of routinely collected data, will generate and analyse both qualitative and quantitative data to achieve the goal of the evaluation. Analysis of qualitative and quantitative data will be carried out using a convergent mixed-methods approach to ensure continuous triangulation of multiple data, which will provide greater understanding of the findings in relation to policy and practice. Broadly, the evaluation could also be considered as process and outcome evaluation.

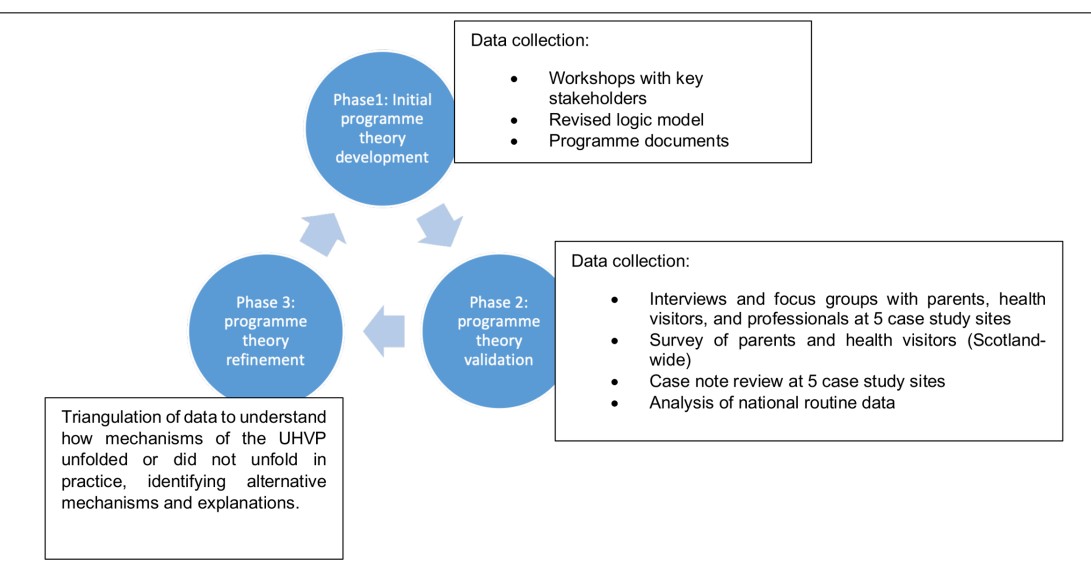

Data collection:
- Workshops with key stakeholders
- Revised logic model
- Programme documents

Data collection:
- Interviews and focus groups with parents, health visitors, and professionals at 5 case study sites
- Survey of parents and health visitors (Scotland-wide)
- Case note review at 5 case study sites
- Analysis of national routine data

Phase1: Initial programme theory development

Phase 3: programme theory refinement

Phase 2: programme theory validation

Triangulation of data to understand how mechanisms of the UHVP unfolded or did not unfold in practice, identifying alternative mechanisms and explanations.

**Figure 1** Conceptual framework and data sources.

## Study setting and population

The evaluation will be conducted over two stages and across all 14 regional health board areas in Scotland. It is expected that the evaluation will take 4.5 years to complete and the recommendations from stage 1 will be implemented by health boards prior to stage 2, which will focus on examining how these recommendations have been implemented, as well as assessing any associated longer term impacts of the UHVP. However, for the purpose of this protocol paper the focus will only be orientated towards stage 1 of the evaluation as much of the methodology outlined will be similar in stage 2.

Although this evaluation is across Scotland, five case study sites, or health board areas, have been selected in order to conduct in-depth case note reviews and qualitative analysis to provide deeper insight into how the UHVP programme works.

## Programme theory
### Initial programme theory development

As part of the implementation of the UHVP, evaluability assessment workshops were conducted in 2016 to review outcomes, identify potential data sources and provide recommendations on the methodology for the evaluation.[13] The process produced a theory of change. This theory of change was designed prior to the full roll-out of the UHVP, and stakeholders involved in producing this agreed that it was important for this to be reviewed further to ensure that the perspectives of national and local stakeholders involved in the planning and implementation of the national roll-out were captured. As such, workshops with national decision makers and health board managers were conducted to review how the implementation processes were carried out, and also examined how they expected the UHVP to work, and in what contexts, in order to produce anticipated outcomes. Data from the workshops were complemented by a review of

key programme documents, which were used to develop the initial programme theory. This initial programme theory informed the rest of the evaluation.

### Programme theory validation

The initial programme theory will be tested to determine how the UHVP was implemented and delivered in practice and to assess how, as well as which outcomes, were achieved. In terms of validating and refining the programme theory, four different elements spanning both qualitative and quantitative methods were identified: qualitative evaluation, case note review, surveys and routine data analysis. As depicted in figure 2, although the evaluation will be Scotland-wide and cover all 14 Scottish health boards, the qualitative and case note review element will only cover five selected health boards or case study sites. The case study sites have been selected based on a self-completion questionnaire that was distributed to Directors of Nursing at each health board in Scotland to enquire about the stage of UHVP implementation. Overall, the main criteria for case study site selection were (1) geographical, population and deprivation factors, (2) UHVP implementation progress and (3) compatibility of local data systems with evaluation outcomes.

### Qualitative research
#### Participants

Interviews and focus group will be conducted with health visitors who deliver the UHVP and parents in receipt of the pathway.

### Health visitors
#### Recruitment and sampling

All health visitors from each study site will be sent information about the evaluation and invited to participate in either a focus group or individual semistructured interview. Individual semistructured interviews will be

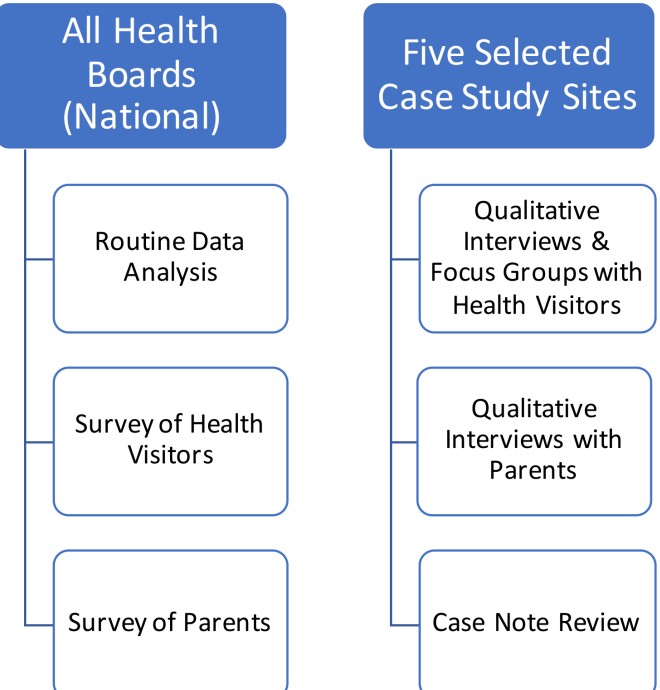

**Figure 2** Overview of study settings and populations in accordance with evaluation components.

conducted with 10 health visitors from each study site in stage 1. A further six participants from each study site will be involved in focus groups (approximately 80 health visitors in total). The mix of individual interviews and focus groups will help to produce rich qualitative data as important issues identified during the individual interviews can be discussed further in focus groups to understand how they are challenged or supported in a group context. This will enable us to draw a more balanced contextual comparisons regarding how different health boards are implementing and delivering the UHVP across Scotland. Focus groups and interviews will last 45 min to an hour and will be audio recorded then transcribed verbatim.

### Parents
The five case study sites will allow for more in-depth examination and understanding of families' experiences of the UHVP. However, given that a number of factors such as, deprivation, rurality, vulnerability or families identified as requiring additional support, might affect parents' experiences of the UHVP, it is important that these differences are captured through purposive sampling. This will provide better understanding of various contextual conditions across subgroups.

In terms of data collection, parents will be recruited via their health visitors, who will inform them about the study and offer them a study information pack. The information pack will include details of how to contact the researchers if they are interested in participating in the study. Potential participants will then be contacted by one of the researchers to arrange a suitable venue and time for interview. The stage 1 topic guide will be informed

by the initial programme theory and it is likely to include questions around families' expectations and experiences of the UHVP, what worked well for them, what did not work and why they thought that was; as well as perceived impact on how the service has influenced their health and well-being. High street store vouchers worth £20 will also be offered to families in return for their time. It is intended that approximately 12 individual semistructured interviews will be conducted within each study site (60 parents overall). Informed consent will be sought to ensure anonymity of all study participants. All interviews will be audio recorded and transcribed verbatim with permission from each participant.

### Qualitative analysis: interviews and focus groups
Qualitative data will be coded and analysed by thematic analysis, using QSR NVivo V.11. Thematic analysis is particularly suited to exploring qualitative data of this nature, as it is possible to examine both within-case and cross-case themes. Three key concepts—context, mechanism and outcome—of realist evaluation will drive the coding process. Two members of the research team will independently analyse the data by selecting the appropriate segments of text and code them appropriately. A final coding frame will be produced after comparison and discussion of the initial coding with the wider research team. The final coding frame will be applied to all data items. Further, similar codes will be grouped together to form overarching themes. The analysis will then advance to the interpretative phase, where potential differences in experience related to implementation, delivery and perceived impact of the UHVP on outcomes will be assessed across participants' characteristics. Particular attention will be paid to how causal mechanisms of the UHVP have been influenced by contextual factors within the study sites to produce intended and unintended outcomes. This will be contextualised in terms of implementation stages for each health board.

### Case note review
The case note review will complement health visitors' perceptions around the practice of implementing the UHVP. To ensure a focus on initial programme theory is maintained, a tool will be created and tested before data collection commences which will not request identifiable personal information in order to maintain anonymity of data gathered.

### Sampling
Within each of the five case study sites health visitor managers will be approached to support the sampling of a selection of current health visitor case notes. Fifteen case notes will be sought from each of the five sites, evenly spread from the regional geographical area, providing a total of 75 case notes for review. As each of the health boards will be at different stages of UHVP implementation, the age ranges of eligible children may vary across each area; however, to ensure that the 75 case notes

**Table 1** Inclusion and exclusion criteria for case note review

| Inclusion criteria | Exclusion criteria |
|---|---|
| Case notes will:<br>► Relate to children who are at least 13 months old at the time of data collection<br>► Initially be selected randomly; however, purposive sampling may be required to ensure that the proportion of vulnerability in those identified is reflective of the geographical area | Case notes will not:<br>► Relate to children who have transferred into the area from another health board area at any time<br>► Relate to children whose parents have participated in the Family Nurse Partnership |

identified will each provide meaningful data for analysis the following criteria, in table 1, will be applied.

## Analysis

Data gathered will focus on the level of adherence to the UHVP identifying: how and when strengths and concerns around well-being were assessed; what interventions were implemented; when and where referrals were made to; and the contribution that health visitors have made to partnership working to support improving outcomes for babies, children and families. The data will be analysed descriptively and compared and contrasted with findings from the other elements of the evaluation.

## A note on COVID-19

It is important to mention that data collection for both the qualitative and case note review elements of the evaluation has just been completed and was not affected by the ongoing COVID-19 pandemic. How COVID-19 has affected the survey and routine data elements have been explained in their respective sections below.

## National survey of health visitors and parents

Two online surveys will be employed, one with parents and one with health visitors. The surveys will provide large-scale, quantitative data on parents' experiences of health visiting and health visitors' experiences and perspectives on the UHVP that are not available through the other elements of the evaluation (eg, routine data or case note review). Phase 1 surveys will serve as a baseline for outcomes achieved by the UHVP as it becomes more embedded in phase 2. However, the change in sampling approach (discussed below) necessitated by COVID-19 limits the extent to which the phase 1 survey can provide a true baseline.

The survey components of the evaluation will be designed around measures relating to the initial programme theory. The surveys will include questions designed to gather the information on how the UHVP is influencing outcomes from the perspective of parents and health visitors. The questions will be designed to measure change, for example, by using scales (eg, strongly agree, tend to agree and so on), rather than binary yes/no

questions. Further information regarding the questions that will be covered in the survey is available in online supplemental file 1. The survey will be conducted across all 14 Scottish health boards in order to gain an overall picture of the implementation of the UHVP.

## Sampling

### Survey of health visitors

Given the relatively small population of health visitors and managers,[14] it is proposed that a census of all health visitors and managers across Scotland will be conducted. A set response rate cannot be guaranteed but based on expert opinion of survey response rates among the public sectors in the UK, it is anticipated that this method could achieve a response rate of approximately 35% for a survey of this type. Based on the total number of health visitors in Scotland, this would estimate around 600 responses to the survey. While it would be desirable to analyse the results of the survey by health board, the extent to which this could be achieved is dependent on the response rate from each health board. In the case that numbers are too low, health boards will be grouped according to a range of factors (eg, stage of roll-out or location) and analysed on that basis.

### Survey of parents

With regard to the parent survey, the sample size will be based on requiring a large enough sample to facilitate disaggregation by, for example, age of child. Originally the parents' survey sample frame was being drawn by local National Health Service (NHS) Boards, however due to the strain placed on the NHS during the COVID-19 pandemic, this approach was deemed no longer viable. However, approval has been gained to use the Scottish Household Survey and Scottish Health Survey (application under consideration) recontact database to contact families with children under 5 years old to invite them to participate. Participation will be entirely voluntary. It is likely that the new approach to sampling may reduce the number of responses we receive.

### Weighting

Corrective weights will likely be applied to compensate for imbalances in the achieved profile of the survey population even if cases where the sample is reasonably representative. This will ensure that when comparing results from the two evaluation stages, the 'same' population will be compared as much as possible. Therefore, the aim of weighting will be to correct for any under-representation or over-representation of different groups of health visitors and parents as a result of non-response.

### Analysis of survey data

A range of statistical techniques, depending on the nature of the data and relevancy to the evaluation will be employed. It is expected that for each question, the proportion of respondents within each subgroup giving a particular response will be analysed. Statistically significant difference between subgroups will be examined,

including difference between health visitors working in health boards with differing stages of implementation, differences by rurality, and deprivation and age of child.

## Routine data analysis

Routine data, captured via health or social work records, or workforce data, will be used to explore the implementation of the UHVP across Scotland, and its impact on relevant outcomes among preschool children. This is a complicated task, due to having to take account of the pathway being introduced in different health boards in various ways, meaning that children in different health boards become exposed at different time points. Routine data items to be explored in both the process and outcome analyses were informed by the logic model, and then refined through discussions with the Scottish Government and the Evaluation Research Advisory Group (see further information about this in the Patient and public involvement section) and explorations around data quality and availability.

Overall, stage 1 data will explore the implementation and outcomes of children prior to the UHVP being introduced, and stage 2 will attempt to capture changes and impact after the introduction of the UHVP. All data will be examined by health boards, and, in addition, where possible, data will be explored by deprivation at Scotland level. Statistical analysis plans will be written for both the process evaluation and the outcome evaluation.

### Process data

The routine data analysis component of the process evaluation will address the following specific research questions:

1. What is the extent to which additional staff have been recruited to health visiting teams to support delivery of the UHVP, and associated changes in indicators of staff well-being such as absence and turnover rates?
2. What is the extent to which the universal child health review elements of the pathway are being delivered, the equity of these contacts and the extent to which contacts vary by health board?
3. What is the extent to which child and family needs are being identified in a timely manner?

Stage 1 process data will capture information about the workforce (eg, staff numbers, turnover, absence rates and students), coverage of the core visits and information about those visits, for example, where they took place, who carried out the review, and which developmental assessment tools were used and how many developmental concerns were raised at different time points. This will be accessed as aggregate data through NHS Public Health Scotland. Data will be described in terms of patterns seen over time and by health boards and Scottish Index of Multiple Deprivation. The same data will be accessed in stage 2 as in stage 1, but at an individual level. This will allow for more detailed analysis of changes since UHVP was implemented in each area, as well as allowing for the calculation of yield of (1) new health plan indicators and

(2) developmental concerns raised following specific reviews, for example. Data will therefore be accessed through the electronic data research and innovation service team by completing an application, which will be assessed by the Public Privacy and Benefit Panel. Data will be accessed remotely through the NHS National Safe Haven.

We will explore data related to the implementation of the UHVP intervention (the process evaluation) using graphical methods.

### Outcome data

The routine data analysis (RDA) component of the outcome evaluation will contribute to this by addressing the following specific research questions:

1. What impact has implementation of the UHVP had on outcomes of children aged up to 3 years relating to:
   a. Parental health-related behaviours?
   b. Child development?
   c. Child physical health?
   d. Child safety?
2. How does impact vary in relation to deprivation?

The logic model items and specific outcome measures used in order to answer the research questions are presented in table 2.

Stage 1 of the outcome RDA will provide baseline data on the listed measures up until implementation of the UHVP. As with the process data at stage 1, this will be accessed as aggregate data through NHS Public Health Scotland. Stage 2 will provide individual participant data for children born on or after the date that the UHVP was implemented in the health board in which those children were born. Data quality will be checked.

The outcome evaluation for the RDA is a natural experiment, assessing the impact of the UHVP on the chosen outcome measures. This involves comparing the occurrence of the outcomes of interest among cohorts of children exposed to the UHVP compared with those that were unexposed to the pathway. We will use an interrupted time series (ITS) design to evaluate the intervention, since this study design is particularly suitable for evaluating the effectiveness of public health interventions. The ITS model consists of data from the unexposed children, data from children who were born on or after the UHVP intervention was implemented (the fully exposed cohort), and the counterfactual, which will be modelled by extrapolating the pre-intervention trend (data from the unexposed cohort) into the post-intervention timeline.

To provide reassurance that our analyses would be able to detect a meaningful and credible level of impact of the UHVP on the outcomes of interest, we undertook formal testing of statistical power to confirm we had adequate (≥90%) power to detect a modest (up to 20% relative) change in the majority of specific outcome measures, based on current data available on the respective outcome measures and numbers of exposed and unexposed children. Only data included in routine data collections can

**Table 2** Outcome measures in the routine data analysis

| Logic model item | Specific outcome measures |
|---|---|
| Improved health behaviours within families (eg, smoke-free homes, breast feeding, weaning and early diet, oral health) | Parental smoking<br>► Primary carer current smoker at 27–30 months<br>► Child exposed to secondhand smoke at 27–30 months<br>Infant feeding<br>► Exclusive breast milk feeding at 6–8 weeks<br>► Any breast milk feeding at 6–8 weeks<br>Immunisations<br>► Complete coverage of universal primary and end infancy immunisations by second birthday<br>Dental care<br>► Any attendance at dentist by second birthday |
| Improved child development and school readiness | Developmental concerns<br>► Any developmental concern at 27–30 months<br>► Any concern about speech, language and communication development at 27–30 months<br>► Any concern about social and emotional development at 27–30 months |
| Improved health outcomes for children and families (eg, healthy child weight, reduced hospital admissions for serious injuries, increase in smoke-free homes, reduced substance misuse) | Child BMI<br>► Child at risk of overweight or obesity (BMI ≥85th centile) at 27–30 months<br>► Child clinically obese (BMI ≥98th centile) at 27–30 months<br>Unintentional injuries<br>► Any hospital admission for unintentional injury by third birthday<br>► Any hospital admission for unintentional poisoning, burn or scald by third birthday<br>► Any hospital admission for unintentional long bone fracture or head injury by third birthday |
| Improved child safety and protection | Child protection interventions<br>► Placed on child protection register at any point between birth and third birthday<br>► Placed on child protection register for ≥6 months between birth and third birthday<br>► 'Looked After Child' status at any point between birth and third birthday<br>► 'Looked After Child' status for ≥6 months between birth and third birthday |
| Reduced inequalities in outcomes and reduced impact of wider inequalities (eg, changing parents approach to parenting despite inequalities) | Examine all outcomes in relation to deprivation (SIMD quintile) |

BMI, body mass index; SIMD, Scottish Index of Multiple Deprivation.

be used which limits the outcomes that can be analysed within this part of the evaluation.

At present, it is unknown what effect COVID-19 will have on data collection and reporting for the latter part of this data collection.

However, the current plan is that phase 1 of the RDA, survey, case review and qualitative data collection and analysis will mostly be completed by December 2020, while these elements in phase 2 will be completed by December 2022.

### Phase 3: programme theory refinement

The phase will involve summarising our refined programme theory based on the findings from phase 2 to robustly articulate how mechanisms of the UHVP unfolded or did not unfold in practice, while identifying alternative mechanisms and explanations.

### IMPLICATIONS OF THE STUDY FINDINGS

The findings of this evaluation will provide an understanding of how the UHVP works and certainly evidence the impact of increased investments in health visiting for children and families across Scotland. Finally, in line with realist evaluation methodology, this evaluation will provide recommendations on how the universal health visiting service can be improved to ensure efficient delivery of the UHVP as well as improving outcomes for children and families, while reducing health inequalities.

## ETHICS AND DISSEMINATION

This protocol has been approved by the School of Health in Social Science Research Ethics Committee, University of Edinburgh. In addition, the case note review has received approval from the Public Benefit and Privacy Panel for health and social care in Scotland. The survey has also received approval from the Scottish Government Public Benefit and Privacy Panel. Approval for the routine data analysis element of the evaluation will be sought from the Public Benefit and Privacy Panel for health and social care in Scotland.

The findings will be prepared as reports to the funders (Scottish Government). The funders are committed to publishing the full report or a summary of the report on their website, which is available to the general public. The findings will also be presented at the annual health visitors' conference in Scotland and other maternal and child health-focused academic conferences. It will also be submitted for publication in peer-reviewed journals.

**Acknowledgements** The authors wish to acknowledge the guidance and contributions of staff at the Scottish Government to the design of the study.

**Contributors** LD, KM, RA, JE, RJ, LM and RW jointly conceived and developed the study proposal. LD led the writing of this manuscript with contributions from KM, RA, JE, MAH, RJ, LM, RO and RW. All authors read and approved the final version of the manuscript.

**Funding** This study is being funded by the Scottish Government (case ref: 362846). At the start of the project, LD and RJ were employed by the Scottish Collaboration for Public Health Research and Policy, which was funded by the Medical Research Council (MR/KO 023209/1) and the Chief Scientist Office.

**Disclaimer** The views expressed in this publication are those of the authors and not necessarily those of the funders.

**Competing interests** None declared.

**Patient and public involvement statement** The study has Evaluation Research Advisory Group. This currently comprises 20 people and includes representatives of the funding organisation, academics, health and care professionals, and members of public organisations around Scotland with interest in maternal and child health. They meet two times per year with the research team to discuss and provide guidance to the various aspects of the project.

**Patient consent for publication** Not required.

**Provenance and peer review** Not commissioned; externally peer reviewed.

of the translations (including but not limited to local regulations, clinical guidelines, terminology, drug names and drug dosages), and is not responsible for any error and/or omissions arising from translation and adaptation or otherwise.

**ORCID iDs**
Lawrence Doi http://orcid.org/0000-0001-6853-5050
Louise Marryat http://orcid.org/0000-0002-6093-4679
Rachael Wood http://orcid.org/0000-0003-4453-623X

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
