## [Reviewer comments · BMJ Open]

ARTICLE DETAILS

TITLE (PROVISIONAL)	Study protocol: A mixed-methods realist evaluation of the Universal Health Visiting Pathway in Scotland
AUTHORS	Doi, Lawrence; Morrison, Kathleen; Astbury, Ruth; Eunson, Jane; Horne, Margaret; Jepson, Ruth; Marryat, Louise; Ormston, Rachel; Wood, Rachael

VERSION 1 – REVIEW

REVIEWER	Dr Ellinor Olander Centre for Maternal and Child Health Research, School of Health Sciences, City, University of London
REVIEW RETURNED	28-Jul-2020

GENERAL COMMENTS	Thank you for submitting this interesting study protocol for publication. National evaluations of health visiting services are rare, and I believe your finished study will provide findings that will be of interest to many. Your study protocol is clearly written and outlines the need for your study well. I have a few very minor suggestions for you to consider. They relate to the clarity of the protocol. - UHVP in the abstract, please either explain this abbreviation or avoid using it altogether.- Please add dates when the different parts of the evaluation started and finished in line with BMJ Open recommendations.- On page 10 you mention the Research Advisory Group, who are part of this group and what is their role within the study?- On page 13 you mention the dissemination of your findings, will findings be disseminated to families at all? If so, how? How will the findings be shared with the health visiting workforce?- On page 15, Patience should be Patient. Also, please provide more detail on your PPI group and their role.- In figure 1, you mention that managers will be interviewed. This is not mentioned in the main manuscript, please clarify if managers will be interviewed as well.
---

REVIEWER	Bethany Boddy Oxford Health NHS Foundation Trust, United Kingdom
REVIEW RETURNED	13-Aug-2020

GENERAL COMMENTS	The study protocol: A mixed method realist evaluation of the universal health visiting pathway in Scotland is well considered and relevant in creating an evidence base for health visiting not just in Scotland but has implications for service delivery across the United Kingdom.
---

	The abstract clearly defines the studies aims in relation to evidencing the impact and outcomes of the enhanced universal health visiting service. Ethics approval is clearly defined, and the limitations and strengths are clearly identified. Background information and evidence base is well set out using recent evidence and identifying gaps in current evidence to identify why the study is needed in the current climate. The complexities of the service delivery and the unique individualised needs assessment health visitors undertake with each family will be explored using several different methods. The combination of interviews, focus groups, and surveying of parents and health visitors offers a broad range of both quantitative and qualitative results that can be analysed to assess impact, outcomes and reflect parents needs and expectations of the service as well as health visitor views and opinions and identifying training needs. The use of a case notes review to compliment health visitor perceptions adds more information for the study results. Consideration of the current COVID 19 pandemic and the effects that this will have on the study and the alterations that may need to be completed are considered and thoroughly explored. Specific outcome measurements in the routine data analysis are clearly set out and relevant to current practice. The study findings aim to explore and provide an understanding of how the universal health visiting pathway works and evidence the impact of how increasing the investment in Scotland for the health visiting service supports families and affects outcomes for children in the first 1001 days of life. The study will also provide information on how to improve health visiting service delivery and explore parents needs and expectations of the service. The study is an opportunity to evidence the work that is done daily by health visitors and measure the often difficult to quantify outcomes, and I feel that there will be important findings and implications for change and improvement for health visiting services through the study when completed.
--	--

VERSION 1 – AUTHOR RESPONSE

Reviewer 1's comments	Response
UHVP in the abstract, please either explain this abbreviation or avoid using it altogether.	Thanks, we have now removed the abbreviation and written UHVP in full as "Universal Health Visiting Pathway".
Please add dates when the different parts of the evaluation started and finished in line with BMJ Open recommendations.	We have added the following in under the methods: Phase 1 of the routine data, survey, case review and qualitative data collection and analysis will mostly be completed by December 2020, whilst these elements in phase 2 will be completed by December 2022.
On page 10 you mention	We have provided further information about the Evaluation Research

the Evaluation Research Advisory Group, who are part of this group and what is their role within the study?	Advisory Group under the Patient and Public involvement section. Below is the information we have included: “The study has Evaluation Research Advisory Group. This currently comprises twenty people and includes representatives of the funding organisation, academics, health and care professionals, and members of public organisations around Scotland with interest in maternal and child health. They meet twice a year with the research team to discuss and provide guidance to the various aspects of the project.”
On page 13 you mention the dissemination of your findings, will findings be disseminated to families at all? If so, how? How will the findings be shared with the health visiting workforce?	We have added the following to the ethics and dissemination section of the manuscript: “The findings will be prepared as reports to the funders (Scottish Government). The funders are committed to publishing the full report or a summary of the report on their website, which is available to the general public. The findings will also be presented at the annual health visitors’ conference in Scotland and other maternal and child health-focused academic conferences. It will also be submitted for publication in peer-reviewed journals.”
On page 15, Patience should be Patient. Also, please provide more detail on your PPI group and their role.	Sorry for the oversight – We have changed ‘patience’ to ‘patient’. As above, we have provided more information about our PPI group as follows: “The study has Evaluation Research Advisory Group. This currently comprises twenty people and includes representatives of the funding organisation, academics, health and care professionals, and members of public organisations around Scotland with interest in maternal and child health. They meet twice a year with the research team to discuss and provide guidance to the various aspects of the project.”
In figure 1, you mention that managers will be interviewed. This is not mentioned in the main manuscript, please clarify if managers will be interviewed as well.	Sorry for the confusion. Managers were involved in Phase 1 workshops and won’t be involved in phase 2 qualitative interviews. We have now made this correction in Figure 1.

VERSION 2 – REVIEW

REVIEWER	Ellinor Olander City, University of London
REVIEW RETURNED	16-Nov-2020
GENERAL COMMENTS	Thank you for revising your manuscript, it reads well and I have no further comments.